# Incidence of Congenital Hypothyroidism Is Increasing in Chile

**DOI:** 10.3390/ijns11030058

**Published:** 2025-07-26

**Authors:** Francisca Grob, Gabriel Cavada, Gabriel Lobo, Susana Valdebenito, Maria Virginia Perez, Gilda Donoso

**Affiliations:** 1Division of Pediatrics, School of Medicine, Pontificia Universidad Católica de Chile, Santiago 8330077, Chile; 2Red de Salud UC CHRISTUS, Santiago 8330033, Chile; 3School of Public Health, University of Chile, Santiago 8350533, Chile; 4Medicina Occidente, San Juan de Dios Hospital, University of Chile, Santiago 8350533, Chile

**Keywords:** congenital hypothyroidism, newborn screening, incidence trends, time-series analysis, thyroid-stimulating hormone, mild CH, Chile

## Abstract

Congenital hypothyroidism (CH) is a leading preventable cause of neurocognitive impairment. Its incidence appears to be rising in several countries. We analysed 27 years of newborn-screening data (1997–2023) from the largest Chilean screening centre, covering 3,225,216 newborns (51.1% of national births), to characterise temporal trends and potential drivers of CH incidence. Annual CH incidence was modelled with Prais–Winsten regression to correct for first-order autocorrelation; additional models assessed trends in gestational age, sex, biochemical markers, and aetiological subtypes. We identified 1550 CH cases, giving a mean incidence of 4.9 per 10,000 live births and a significant yearly increase of 0.067 per 10,000 (95 % CI 0.037–0.098; *p* < 0.001). Mild cases (confirmation TSH < 20 mU/L) rose (+0.89 percentage points per year; *p* = 0.002). The program’s recall was low (0.05%). Over time, screening and diagnostic TSH values declined, total and free T4 concentrations rose, gestational age at diagnosis fell, and a shift from thyroid ectopy toward hypoplasia emerged; no regional differences were detected. The sustained increase in CH incidence, alongside falling TSH thresholds and growing detection of in situ glands, suggests enhanced recognition of milder disease. Ongoing surveillance should integrate environmental, iodine-nutrition, and genetic factors to clarify the causes of this trend.

## 1. Introduction

Newborn screening (NBS) for congenital hypothyroidism (CH) is a critical public health intervention aimed at preventing intellectual disability associated with untreated CH. Early diagnosis and prompt initiation of levothyroxine therapy can lead to normal neurocognitive development into adulthood [1].

The global prevalence of CH has increased since 1969 [2,3,4,5,6], a trend that may reflect the widespread adoption of NBS programs and the lowering of TSH diagnostic thresholds [4,7,8]. Additionally, the profile of newborns screened for primary CH in developed nations has undergone significant changes with an increase in preterm and multiple births, and infants admitted to NICU [9,10]. Further contributing factors may include iodine deficiency [11] and ethnic differences [12].

In Chile, the NBS program for CH began as a pilot initiative in 1984, led by the National Institute of Nutrition and Food Technology (INTA) and the Ministry of Health. In 1992, the National Program for Mass Screening of Phenylketonuria and CH was officially launched and gradually expanded to achieve nationwide coverage by 1998 [13]. Currently, the NBS program covers 99% of newborns. This high coverage has proven effective in ensuring normal neurocognitive outcomes in children diagnosed with CH [14]. To date, however, there are no published data on the incidence or etiological distribution of CH in the Chilean population. The objective of this study was to analyse temporal trends in the incidence of CH in Chile between 1997 and 2023.

## 2. Materials and Methods

The Chilean NBS program has always been based on the measurement of neonatal TSH. Two NBS centres in Chile receive the special blood collection paper cards from the babies born in the public health system (Centres 1 and 2), representing around of 80% of the country’s births, and process 76% and 24% of the samples coming from 16 regions of the nation, respectively [13]. Whole-blood TSH on dried blood samples is collected on special blood collection paper. According to national guidelines, the timing of sample collection is based on gestational age: in term infants (≥37 weeks), the sample is obtained between 40 and 48 hours of life; in late preterm infants (36–36⁶/₇ weeks), the sample is collected on the 7th day of life; and in preterm infants (≤35 weeks), two samples are collected—one at 7 days and another at 15 days of life. These time intervals have remained unchanged throughout the entire duration of the screening program. Information related to CH diagnoses is collected, including sex, age at sampling, screening and confirmatory TSH levels, and thyroid scintigraphy results when available. Data on gestational age has been systematically recorded since 2009.

Between 1998 and 2006, TSH screening was performed using IRMA DPC (Immunoradiometric Assay, CLINITEST, Diagnostic Products Corporation, Los Angeles, CA, USA), and in 2006 switched to the Delfia^®^ time-resolved fluorometry system (PerkinElmer) (Waltham, MA, USA). In May 2018, the laboratory adopted the fully automated GSP platform (PerkinElmer) (Waltham, MA, USA). To harmonise results after transitioning from AutoDelfia to GSP platforms, a re-evaluation of TSH cutoff values was performed using ROC curve analysis on a dataset of 55,717 newborns screened between May and October 2018. This analysis, identified a new TSH action cutoff of ≥13 µIU/mL (serum-equivalent), compared to the previous threshold of 15 µIU/mL.

Under the Delfia method (Figure 1), an initial TSH measurement < 13 µIU/mL is deemed screen-negative; values ≥ 13 µIU/mL trigger two repeat assays on separate card punches, with the mean of all three readings dictating next steps: <14 µIU/mL is negative, 14.0–23.9 µIU/mL prompts a second-card retest, and ≥24.0 µIU/mL leads directly to confirmatory serum TSH/free T4. If the second card mean is ≥15 µIU/mL, the newborn is referred; otherwise, no action is taken.

With the GSP platform (Figure 2), samples with initial TSH ≥ 10 µIU/mL undergo duplicate reanalysis: if either repeat is ≥13 µIU/mL, referral for confirmatory testing occurs; if both are <13 µIU/mL, the screen is negative. 

Throughout the study period, TSH concentrations were reported as serum-equivalent values, using a standardised correction factor based on an assumed haematocrit of 55% (1 µU/mL in whole blood = 2.22 µU/mL in serum).

A diagnosis of CH is confirmed when confirmatory sample serum TSH is >10 mU/L and total T_4_ is <10 µg/dL. In cases where free T_4_ (FT_4_) levels are within the reference range of the specific assay employed by each centre, a diagnosis of hyperthyrotropinemia (HT) is made. When available, thyroid scintigraphy using 99mTc is performed to determine the underlying aetiology.

A retrospective review was conducted using data from the national NBS program, including infants born within the Chilean public health system between 1997 and 2023, whose special blood collection paper cards were processed at Centre 1. This center receives samples from 11 of Chile’s 16 regions, representing approximately 51.1% of all newborns in the country.

The recall rate was defined as the proportion of newborns recalled for a repeat sample or confirmatory test—owing to elevated TSH, inadequate sample, or analytical interference—relative to all newborns with a valid initial specimen. It was calculated as: Recall rate (%) = (Recalled newborns/Total newborns screened) × 100, with exact 95% confidence intervals derived by the Clopper–Pearson method.

### Statistical Analysis

Time-trend analyses were performed to describe changes in CH incidence and their potential determinants from 1997 to 2021. Data from 1992 to 1996 were excluded because nationwide screening was still in its pilot phase and lacked full population coverage. First-order autocorrelation in the yearly series was handled with Prais–Winsten regression. The primary outcome was the annual incidence of confirmed CH cases.

Time trends in the incidence of CH were evaluated using Prais–Winsten regression models, with calendar year as the independent variable. In addition to analyzing overall incidence, additional models assessed changes over time in gestational age, sex distribution, age at diagnosis, and hormone concentrations, including screening (whole-blood) and confirmatory (serum) TSH, as well as total and free T4 levels. To investigate whether the observed increase in incidence was driven by mild cases, we conducted two separate analyses: (i) modeling the annual incidence of mild CH, defined by a screening TSH concentration < 20 µUI/mL, and (ii) modeling the yearly proportion of mild cases among all confirmed diagnoses. Conversely, to examine trends in more severe CH, we calculated the annual number of cases with screening TSH ≥ 20 µUI/mL and applied a linear regression model to determine whether the frequency of these non-mild cases changed over time. Finally, additional Prais–Winsten models were applied to assess trends in the distribution of aetiologic subtypes (e.g., ectopy, athyreosis, hypoplasia). 

For each administrative region, annual incidence was calculated as Incidence = Confirmed CH cases × 10,000. Regional heterogeneity was assessed with a mixed-effects linear model that treated region as a fixed effect and year as a random intercept, thereby accounting for clustering over time.

Sex-specific differences in CH aetiology were examined among cases with definitive scintigraphic classifications (ectopy, athyreosis, hypoplasia, goitre, normal scan) using Fisher’s exact test; cases with missing sex were excluded. All analyses were conducted on yearly aggregated data in Stata 17 (StataCorp, College Station, TX, USA), with statistical significance set at *p* < 0.05.

## 3. Results

### 3.1. Overall Incidence (1997–2023)

Between 1997 and 2023, 3,225,216 newborns (51.1% of Chilean live births) were screened for congenital hypothyroidism (CH) at Centre 1. A total of 1550 cases were confirmed, yielding a mean incidence of 4.9 per 10,000 live births, rising from 3.91 in 1997 to 5.75 in 2023 (Figure 3).

### 3.2. Diagnostic Characteristics

Sixty per cent of affected infants were female. The median TSH value at screening was 117.6 µUI/mL (IQR: 65.7–172.0), and at confirmatory serum testing it was 99.2 µUI/mL (IQR: 59.4–100.0). The median total T4 concentration was 5.43 µg/dL (IQR: 4.25–8.00), and the median free T4 concentration was 0.90 ng/dL (IQR: 0.78–1.07). Scintigraphy was not obtained in 58.8 % of cases, which were classified as “unknown” aetiology. Among the remainder, ectopy was most frequent, followed by athyreosis, goitre and hypoplasia (14.8%, 8.4%, 9% and 6%, respectively; Table 1).

### 3.3. Diagnostic Performance

The recall rate was 0.05% (95% CI 4.7–5.2 per 10,000). Of the 1604 screen-positive newborns, 10 lacked complete follow-up and were excluded from performance metrics. The overall positive predictive value (PPV) was 32% (95% CI 30–35%). Stratified by severity, the PPV for severe CH was 58% (95% CI 53–63%), while the PPV for mild CH was 14% (95% CI 12–16%). For comparison, the recall rate during the RIA period was 0.047% (95% CI: 3.6–6.2 per 10,000), indicating similar program performance over time.

### 3.4. Time-Trend Analyses

Prais–Winsten autoregressive regression demonstrated a significant annual increase of 0.067 cases per 10,000 live births (95% CI 0.037–0.098; *p* < 0.001; R^2^ = 0.55). Mild cases (confirmation TSH < 20 mUI/L) rose by 0.069 per 10,000 per year (95% CI 0.028–0.109; *p* = 0.002; R^2^ = 0.29) (Figure 4), and their share of all CH diagnoses increased by 0.89 percentage points annually (95% CI 0.37–1.40; *p* = 0.002; R^2^ = 0.31). Severe cases (screening TSH ≥ 20 µUI/mL) decreased significantly over time (β = −0.015; *p* = 0.017), suggesting a decline in the incidence of clinically overt CH.

### 3.5. Regional Variation

Incidence by region of the country (cases per 10,000 = [confirmed CH/newborns screened] × 10,000) showed no significant differences in a mixed-effects model with region as a fixed effect and year as a random effect (all *p* > 0.05).

### 3.6. Aetiological Trends and Sex Distribution

The proportion of ectopy and athyreosis decreased (β = −0.007, *p* = 0.019 and −0.0027, 0.092, respectively) whereas hypoplasia increased (β = 0.0037; all *p* ≤ 0.050). Ectopy showed strong female predominance (72% female, *p* < 0.001), while hypoplasia, goitre and normal scans were more common in males (61%, 59%, and 72% male; *p* = 0.005, 0.019, <0.001, respectively). Dysgenesis (ectopy + athyreosis) had a female/male ratio of 2.47, versus 0.79 for other aetiologies (*p* < 0.001).

### 3.7. Biochemical Trends

Screening TSH and diagnostic serum TSH both declined over time (β = −8.54 µUI/mL and −3.93 mU/L per year; *p* < 0.001), whereas total T4 and free T4 rose (β = 0.25 and 0.03 µg/dL per year; *p* ≤ 0.002).

### 3.8. Multivariable Analysis and Prematurity

In a multivariable Prais–Winsten model including gestational age, sex, aetiology and hormone concentrations, none independently explained the secular rise in incidence. Prematurity (data available 2009–2023) increased non-significantly over time (β = 0.0098; *p* = 0.13) and was not associated with CH incidence (β = 3.97; *p* = 0.31).

## 4. Discussion

This study demonstrates a significant upward trend in the diagnostic incidence of CH in Chile over the past two decades. A consistent increase of approximately 0.07 cases per 10,000 live births annually (*p* < 0.001) was observed. This finding aligns with reports from other countries that have documented rising CH incidence over time. In Ireland, the incidence of CH has significantly increased over the past 37 years. with the rate rising from 0.27 cases per 1000 live births between 1979 and 1991 to 0.65 cases per 1000 live births between 2005 and 2016 [15]. Similarly, in China, the incidence increased from 4.01 per 10,000 births in 2012 to 5.77 per 10,000 births in 2019. with factors such as changes in TSH cutoff values and increasing preterm birth rates contributing to this trend [16]. Northern Ireland has also seen a near tripling of CH incidence over 40 years, from 26 cases per 100,000 live births in 1981 to 71 cases per 100,000 in 2019 [17]. In France, the incidence of permanent CH in females increased by 8.9 per year from 2014 to 2019 [18]. A global meta-analysis reported a 52 increase in the prevalence of CH from 1969 to 2020, suggesting that factors such as neonatal screening and changes in diagnostic criteria may be contributing to the rise [2]. However, Finland presents a contrasting picture, where the incidence of CH has remained stable over a 24-year period, indicating that regional differences exist [19].

Several hypotheses have been proposed to explain the increasing incidence of CH worldwide, including improvements in screening sensitivity, changes in diagnostic thresholds [4], and demographic changes [20]. Notably, the increase has been largely attributed to the detection of milder cases, particularly those with in situ thyroid glands. which may not have been diagnosed earlier under more stringent screening criteria [21]. In our analysis, we observed declining trends in both screening and serum TSH concentrations, along with increases in total and free T4 levels. These trends may reflect enhanced detection of milder forms of CH. Additionally, ectopy declined significantly (β = −0.007; *p* = 0.012), whereas the reduction in athyreosis was not significant (β = −0.0027; *p* = 0.084). This trend supports the hypothesis that the increasing incidence of CH may be partly driven by the detection of milder forms of the disease, as previously reported. For instance, in Québec, lowering the TSH cutoff in screening tests led to an increase in the detection of cases with normal-size glands in situ and unknown aetiology, while the incidence of dysgenesis and goitre remained stable [6]. Additionally, the implementation of the GSP platform in 2018 may have increased the detection of milder or borderline cases, contributing to the upward trend in incidence.

Furthermore, the observed decrease in gestational age at approximately one day earlier per year—suggests that an increasing number of cases might being detected among preterm infants. The potential role of prematurity was explored. While the proportion of preterm births appeared to rise slightly, this trend was not statistically significant, and prematurity did not explain the increase in CH incidence in our population. Several studies have highlighted the elevated risk of CH in preterm infants due to their increased survival rates [8] and their unique and dynamic pattern of thyroid hormone levels, explained by the immaturity of the hypothalamic–pituitary–thyroid (HPT) axis; the withdrawal of maternal thyroxine (T4) after birth, exposure to iodine especially in iodine deficient areas; medications; birth weight, and the persistence of fetal metabolism [22]. In China. the increasing rate of preterm births has been identified as a contributing factor to the rising incidence of CH, although the contribution is considered limited compared to other factors like changes in screening practices [16]. In our cohort, 11.7% of confirmed CH cases occurred in infants born at ≤35 weeks and 6.4% in those born at 36–36⁶/₇ weeks. Given that preterm births account for approximately 8–10% of all live births in Chile, these figures suggest a disproportionately higher incidence of CH among preterm infants. However, because our database lacks precise denominators for the total number of preterm births screened each year, we cannot calculate exact incidence rates for this subgroup. This limitation underscores the need for dedicated data collection on gestational age to more accurately assess the contribution of prematurity to CH incidence.

Our findings confirm a clear sex-based difference in the etiological distribution of thyroid dysgenesis, with a pronounced female predominance. This observation is consistent with prior studies suggesting sex-linked susceptibility or inherent developmental differences influencing thyroid morphogenesis and migration [23]. Although the overall proportion of female patients showed a slight but statistically significant decline over time, this trend alone is unlikely to account for the observed increase in CH incidence. Notably, a recent report from France described a rising incidence of permanent CH among females [18], supporting the notion that temporal and regional factors may influence sex-specific trends. Concurrently, we observed a significant increase in cases attributed to thyroid hypoplasia, alongside a decrease in those classified as ectopy or athyreosis. In contrast, dyshormonogenetic presentations (hypoplasia, goitre, and normal scintigraphy) disproportionately affected males, indicating that sex-specific factors—potentially hormonal, placental, or genetic—together with changes in population admixture, may be driving the upward trend in milder CH forms among boys. Unfortunately, because the national screening program collects data only at diagnosis and does not include systematic follow-up, we cannot distinguish whether these milder cases are transient or permanent.

Despite evaluating multiple clinical and biochemical variables, none of the factors examined fully accounted for the upward trend in incidence, suggesting that additional, unmeasured contributors may be at play. These could include changes in iodine status, maternal health conditions, or unidentified environmental factors [24], which underscores the need for future research integrating environmental, nutritional, and genetic data.

## 5. Conclusions

In summary, we observed a sustained increase in the incidence of CH in Chile, which was associated with a decline in both screening and diagnostic TSH concentration, as well as an increase in cases with in situ thyroid glands. This pattern may reflect a rise in incidence driven by the detection of milder forms of CH. Overall, these findings underscore the importance of continued monitoring of CH incidence and its determinants. Future research should investigate environmental exposures, regional variation in iodine sufficiency, and potential genetic contributions.

## Figures and Tables

**Figure 1 IJNS-11-00058-f001:**
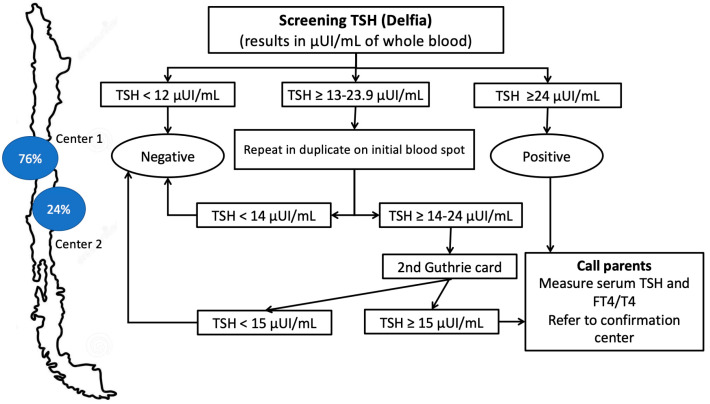
Screening algorithm for CH in Chile (Delfia^®^ time-resolved fluorometry system, 2006–2018).

**Figure 2 IJNS-11-00058-f002:**
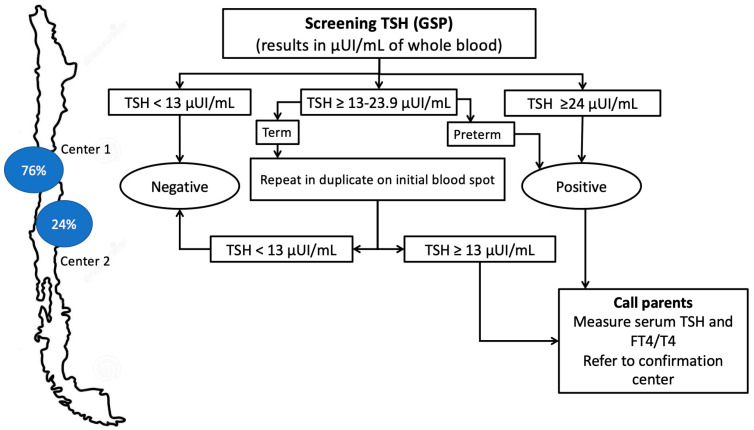
Screening algorithm for CH in Chile (GSP platform, 2018–2023).

**Figure 3 IJNS-11-00058-f003:**
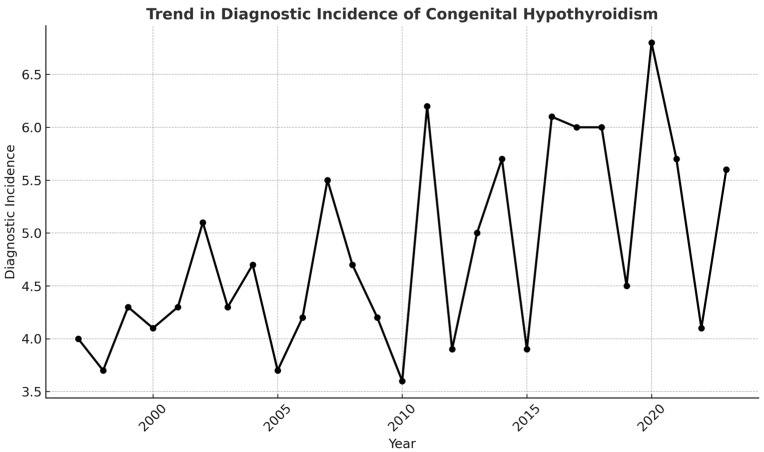
Overall annual incidence rate of CH per 10,000 live births (Chile, 1997–2023).

**Figure 4 IJNS-11-00058-f004:**
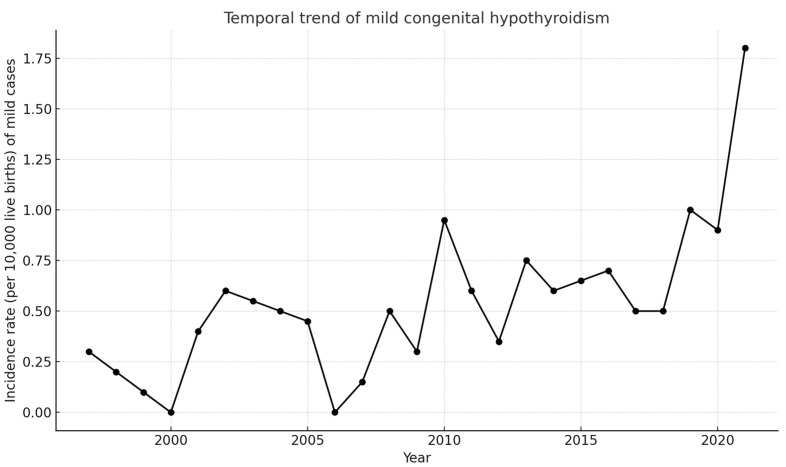
Annual incidence rate of mild cases of CH per 10,000 live births (Chile, 1997–2023).

**Table 1 IJNS-11-00058-t001:** Number of CH cases by aetiology and overall incidence rates per year, Chile, 1997–2023.

Year	Screened Newborns (*n*)	CH Cases (*n*)	Incidence Rate (per 10,000 Live Births) (Mean)	Ectopy (*n*,%)	Athyreosis (*n*,%)	Hypoplasia (*n*,%)	Goitre (*n*,%)	Normal (*n*,%)	Unknown (*n*,%)
1997	125,245	49	3.91	14 (28)	3 (6)	1 (3)	12 (25)	0 (0)	19 (39)
1998	145,086	53	3.65	15 (28)	13 (25)	4 (8)	3 (6)	0 (0)	17 (33)
1999	141,139	62	4.39	20 (33)	7 (12)	3 (5)	4 (7)	1 (2)	25 (40)
2000	141,093	57	4.04	15 (27)	5 (9)	1 (2)	13 (22)	2 (4)	21 (36)
2001	138,369	61	4.41	12 (20)	6 (10)	4 (6)	9 (14)	1 (2)	29 (48)
2002	137,282	69	5.03	14 (21)	5 (7)	5 (7)	11 (16)	0 (0)	34 (49)
2003	130,707	57	4.36	9 (15)	10 (17)	1 (2)	2 (4)	2 (4)	32 (57)
2004	123,565	58	4.69	5 (9)	5 (9)	0 (0)	3 (5)	0 (0)	45 (77)
2005	120,896	45	3.72	1 (3)	1 (3)	0 (0)	0 (0)	0 (0)	42 (94)
2006	118,802	50	4.21	6 (11)	2 (3)	2 (3)	2 (5)	2 (3)	38 (76)
2007	122,408	68	5.56	4 (6)	0 (0)	1 (2)	1 (2)	0 (0)	61 (90)
2008	123,392	58	4.7	11 (19)	11 (19)	3 (6)	5 (9)	2 (4)	26 (44)
2009	125,383	52	4.15	6 (12)	1 (2)	3 (5)	5 (10)	0 (0)	37 (71)
2010	126,444	44	3.48	10 (22)	2 (5)	3 (7)	4 (9)	3 (7)	22 (49)
2011	121,420	75	6.18	10 (14)	9 (12)	5 (7)	7 (9)	2 (3)	41 (55)
2012	118,081	52	4.4	3 (5)	4 (8)	3 (6)	6 (12)	2 (4)	34 (65)
2013	116,496	58	4.98	7 (12)	7 (12)	1 (2)	3 (6)	1 (2)	39 (67)
2014	119,336	68	5.7	7 (11)	8 (12)	4 (6)	4 (6)	0 (0)	44 (65)
2015	115,322	74	6.42	12 (16)	6 (8)	4 (5)	7 (10)	0 (0)	45 (61)
2016	108,394	43	3.97	5 (12)	1 (2)	1 (2)	3 (7)	0 (0)	33 (76)
2017	108,911	66	6.06	5 (7)	5 (7)	11 (16)	4 (6)	1 (2)	40 (60)
2018	120,561	73	6.06	9 (12)	3 (4)	8 (11)	9 (12)	3 (4)	41 (56)
2019	113,095	53	4.69	5 (9)	4 (7)	7 (14)	5 (10)	5 (10)	26 (49)
2020	96,040	66	6.87	7 (10)	3 (5)	9 (13)	5 (7)	5 (8)	37 (56)
2021	83,934	49	5.84	4 (9)	3 (6)	6 (13)	3 (7)	3 (7)	28 (58)
2022	93,355	38	4.07						
2023	90,460	52	5.75						
Total	3,225,216	1550	4.9	216 (14.8)	124 (8.5)	90 (6.2)	130 (8.9)	35 (4.5)	856 (58.6)

## Data Availability

The de-identified individual-level dataset generated during this study is available from the corresponding author on reasonable request. Aggregated incidence data and the Stata code used for analysis are also available on request.

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
