# Peer review of "Incidence of Congenital Hypothyroidism Is Increasing in Chile"

_2409-515X, 2025, doi:10.3390/ijns11030058_

Round 1
Reviewer 1 Report
Comments and Suggestions for Authors
See attached file

Author Response
Reviewer 1
General Comments:
We appreciate the reviewer’s positive assessment of our work and their thoughtful suggestions to strengthen the manuscript.
Specific Comments
Comment 1:
The authors state that dried blood was collected on filter paper after 40 hours of life but that CH screening occurred at 7 days or 7 & 15 days depending on gestational age. What testing if any is conducted between 40 hours and 7 days of age?
Response 1:
Thank you for this observation. We have clarified the protocol in the Methods section (Page 2, Line 82). In Chile, newborn screening for congenital hypothyroidism follows national guidelines that determine sample timing based on gestational age: No testing is conducted between 40 and 7 days of age.
- For term infants (≥37 weeks gestational age), the sample is collected between 40–48 hours of life.
- For late preterm infants (36–36⁶/₇ weeks), the sample is collected at 7 days of life.
- For preterm infants (≤35 weeks), two samples are collected: the first at 7 days and the second at 15 days of life.
Comment 2:
What is the process for premature babies? If the CH screening test is positive at 7 days are babies referred and only those with a normal result retested at 15 days? I presume the retesting is to ensure babies are not missed due to delay in the TSH surge due to prematurity but this is not stated.
Response 2:
Thank you for this question. Under both the Delfia and GSP screening protocols, premature infants follow the same overall workflow as term infants; however, sampling timing differs.
Delfia protocol: Preterm infants (≤35 weeks’ gestation) have their first dried‐blood‐spot sample collected at 7 days of life and a second at 15 days of life. Each sample is processed and interpreted according to the standard Delfia algorithm (initial measurement, duplicate reanalysis if ≥13 µIU/mL, etc.).
GSP protocol: For preterm infants, any initial screening TSH ≥ 13 µIU/mL on the first sample at day 7 triggers an immediate request for confirmatory serum testing (TSH and/or free T4), rather than proceeding to duplicate punches. If the first sample is < 13 µIU/mL, a second dried‐spot sample at day 15 undergoes duplicate reanalysis (analytical cut‐off ≥ 10 µIU/mL; action cut‐off ≥ 13 µIU/mL) in the same manner as term infants.
We have added this clarification to the Methods section (Page 2, Line 78).
Comment 3:
It is stated that TSH concentrations are reported as serum equivalent units but the formula used for hematocrit correction is not stated. Presumably a common correction formula is used for all samples as it would not be possible to assess hematocrit in the bloodspots?
Response 3:
Thank you for this clarification request. We have added the conversion formula used for serum equivalence in the revised manuscript (Page 3, Line 104). A standardized correction factor is applied, assuming an average hematocrit of 55%, whereby: 1 µU/mL in whole blood = 2.22 µU/mL in serum. This conversion allows for uniform interpretation of results without direct hematocrit measurement in dried blood spots.
Comment 4:
Please can the authors confirm that the TSH levels quoted throughout the paper are all corrected values? Authors state that corrected values were introduced in 2006 with implementation of the Delfia method – if this is the case I would question the appropriateness of combining uncorrected data prior to 2006 with corrected values post 2006.
Response 4:
Thank you for this observation. We confirm that throughout the entire study period, TSH values have been reported as serum-equivalent concentrations. Even prior to 2006, when the IRMA DPC method was in use, a standardized conversion factor was applied to express results in serum units. Therefore, the data presented in the manuscript are comparable over time. This has now been clarified in the Methods section (Page 3, Line 87)
Comment 5:
Is diagnostic confirmatory testing based on TSH and total T4 or TSH and free T4? The paper mentions a cut-off for total T4 but then states that when free T4 levels are normal a diagnosis of hyperthyrotropinemia (HT) is made.
Response 5:
Thank you for your observation. Confirmatory testing includes TSH and either total T4 or free T4, depending on the resources available at the referral center. For total T4, a cut-off of 10 µg/dL is used. For free T4, the interpretation is based on the reference range of the specific assay employed by each center. A diagnosis of hyperthyrotropinemia is made when free T4 is within the normal range for the corresponding method. This clarification has been added to the Methods section (Page 3, Line 109).
Comment 6:
The text and Figure 1 are inconsistent with respect to borderline repeats. The text states that if the repeated value is below 15 no further action is taken whereas the figure states referral for diagnostic confirmation if the repeat TSH is >10. Further are these serum corrected levels? If so a serum corrected level of 10 (equivalent approx. to a measured level of 5 in the bloodspot) would be an extremely low threshold for referral.
Response 6:
Thank you for the opportunity to clarify. All TSH values in our manuscript are reported as serum‐equivalent concentrations, derived from whole‐blood spots. We have now clearly specified in the Methods which platforms were used (Delfia vs. GSP), the precise thresholds for term and preterm infants, and the two‐step cut‐off strategy.
In the Delfia algorithm, initial TSH values ≥ 13 µIU/mL trigger duplicate reanalysis. If both repeats are < 15 µIU/mL, the screen is negative; if either repeat is ≥ 15 µIU/mL, the newborn is referred for confirmatory serum testing.
In the GSP algorithm, an initial TSH ≥ 10 µIU/mL (analytical cut‐off) leads to duplicate reanalysis; an average ≥ 13 µIU/mL (action cut‐off) results in referral.
To ensure full consistency, we have updated Figure 1 to reflect the corrected Delfia thresholds and added Figure 2 illustrating the GSP workflow. Please refer to these figures for the detailed decision pathways.
Comment 7:
Statistical analysis – I confess to not being familiar with the Clopper-Pearson method of deriving confidence intervals or with Prais-Winsten regression modelling so cannot comment on whether these are best statistical tools to use for this particular study.
Response 7:
Thank you for this comment. The Clopper-Pearson method is a widely used exact method for calculating confidence intervals for binomial proportions, ensuring conservative estimates. Prais-Winsten regression is suitable for time-series data with autocorrelation, which applies to our dataset.
Comment 8:
Table 1 – it would be useful to see the combined figures for athyreosis and hypoplasia vs the combined figures for normal or enlarged glands as this would illustrate whether there is a likely increase in thyroid dyshormonogenesis versus thyroid dysgenesis as has been reported in other studies.
Response 8:
We thank the reviewer for this insightful suggestion. However, we believe it is more appropriate to maintain the current structure of Table 1, as this format is consistent with how other studies have reported congenital hypothyroidism incidence according to distinct scintigraphic categories. Additionally, to explore the reviewer’s point regarding the potential increase in milder forms associated with dyshormonogenesis, we conducted a separate time-trend analysis focused on cases with screening TSH <20 µU/mL. This analysis provides meaningful insight into the observed rise in milder cases over time and is presented in the Results section (Page 6, Line 170).
Comment 9:
Can the authors confirm that the same cut-offs were used for TSH throughout the time period of this study? Figure 3 appears to show a steady rise in mild cases of CH up to approx. 2017/18 and then a sharper rise – was this associated with any change to the screening programme or TSH cut-offs?
Response 9:
Thank you for this observation. Throughout the study period, TSH screening platforms and cut-off values have been standardized to ensure continuity of results:
- 1998–2006: Screening was performed using IRMA DPC (CLINITEST). All results were converted to serum-equivalent units (1 µU/mL whole blood = 2.22 µU/mL serum) and reported against a cut-off of 20 µU/mL in whole blood.
- 2006: The Delfia® time-resolved fluorometry system (PerkinElmer) was introduced. A method-comparison exercise showed that a 15 µU/mL whole-blood cut-off on Delfia produced equivalent clinical sensitivity and specificity to the previous 20 µU/mL threshold on IRMA DPC; hence, the action cut-off was lowered to 15 µU/mL from that point onward Informe valores de cort….
- May 2018: The laboratory replaced the AutoDelfia platform with the GSP® Genetic Screening Processor (PerkinElmer).
- August 1, 2019: Following a population‐based ROC analysis of 55 717 samples, a two-tier strategy was implemented: an analytical cut-off of ≥ 10 µU/mL and an action cut-off of ≥ 13 µU/mL in whole blood
We have now detailed these platform transitions and cut-off harmonizations in the Methods section (Page 3, line 87). Additionally, we note in the Discussion that the sharper rise in mild CH cases after 2017–18 may in part reflect the 2019 algorithm update on the GSP platform (Page 8, line 235).
Comment 10:
There is no mention of the potential impact of any increase in cases of gland in-situ (associated with dyshormonogenesis, which is for the most part autosomal recessively inherited) versus athyreosis (approx. 95% sporadic) on the gender distribution.
Response 10:
Thank you for this comment. In our cohort, the proportion of ectopy decreased significantly over time (β = –0.007; p = 0.012), whereas the decline in athyreosis did not reach statistical significance (β = –0.0027; p = 0.084). Sporadic thyroid dysgenesis is well‐recognized to exhibit a female predominance (female:male ratio ~2:1; PMID: 11344199), a pattern we also observed: dysgenesis (ectopy + athyreosis) had a female:male ratio of 2.47. Nevertheless, conditions suggestive of dyshormonogenesis—hypoplasia, goitre, and normal scans—were significantly more common in males (61%, 59%, and 72% male, respectively). Although consanguinity in Chile is historically low (PMID: 9334449), recent migratory changes may influence population genetics in ways that remain unexplored. Taken together, these findings suggest that both sex-specific biology and evolving population structure may underlie the disparate trends in thyroid aetiologies. (Page 9, Line 279).
Comment 11:
Did the authors look at whether the milder cases of CH were associated with an increase in transient vs permanent CH? If not, this may be worthy of mention.
Response 11:
Thank you for this relevant point. Unfortunately, the national screening program collects data only at the time of diagnosis and does not systematically capture follow-up information. Therefore, we cannot distinguish between transient and permanent CH in our dataset. This limitation and the need for follow-up studies have been noted in the Discussion section (Page 9, Line 275).
Reviewer 2 Report
Comments and Suggestions for Authors
Thank-you for this nice presentation of the rising incidence of congenital hypothyroidism in Chile. Specific comments are:
Introduction ll38-55 landscape of screening for CH in the world and LMICs isn’t relevant to changing incidence in Chile – suggest delete
L77 please do not call blood collection paper filter paper. Although this term is widely used, it isn’t filter paper it is special blood collection paper.
Has there been any change in the age at which samples are collected over the study period? Worldwide there is a trend to collecting samples earlier. If samples are collected earlier, and the cutoff remains the same it would be expected that the recall rate would increase with consequent detection of more mild cases.
Ll82-84 please comment on what if any difference there is in analytical values provided by DPC and Delfia assays (if the direct comparison wasn’t done at the time of changeover the CDC programme data provides a useful comparison). Did the positive test rate (recall rate) change when the assay changed and the cutoff remained the same?
L86 please explain how whole blood values were corrected for hematocrit given the big differences in hematocrit observed in premature infants compared to term newborns.
L87 earlier you say values were reported in serum equivalents and in this sentence whole blood levels over 15mIU/L – is this a serum equivalent or whole blood?
L90 ‘if it remains’ does this refer to the average value of the two repeats, the median value of the 3 results, the highest value or something else?
L137 please use appropriate statistical descriptions of data eg 128+-73 implies a normal distribution with 96% of results lying between -18 and 274 – neither of these statements is likely to be true.
L143 recall rate is 0.05 please give a unit eg % (rate is number per a denominator). What was the positive predictive value for severe and mild cases?
L189 the incidence of CH in premature infants is higher than that in term babies please do you have a separate incidence for this population – although I agree the relative number is small so an increase of prematurity is unlikely to be the cause of the population increase.
If there has not been a change in screening practice (changed assay or earlier collection increasing the recall rate) the increase may be due to a change in population (eg Albert et al J Clin Endocrinol Metab 97, 3155-60, 2012) or an environmental factor (eg Chamot Journal of Hazardous Materials Advances vol17 Feb 2025, Street et al Front Endocrinol vol15 Jun 24). Do you have a hypothesis as to the cause of the observed increase?
Author Response
Reviewer 2
Comment 1:
Introduction, lines 38–55: The landscape of screening for CH in the world and LMICs isn’t relevant to changing incidence in Chile – suggest delete.
Response 1:
We agree with the reviewer and have removed this section to improve focus and clarity.
Comment 2:
L77: Please do not call blood collection paper "filter paper." Although widely used, it isn’t filter paper—it is specialized blood collection paper.
Response 2:
Thank you for this correction. We have replaced "filter paper" with "specialized blood collection paper" in the text (Page 2, line 77).
Comment 3:
Has there been any change in the age at which samples are collected over the study period? Earlier collection with unchanged cut-offs could increase mild case detection.
Response 3:
Thank you for this observation. No changes have occurred in the timing of sample collection during the study period. This has been clarified in the Methods section (Page 2, Line 82).
Comment 4:
L82–84: Please comment on analytical differences between DPC and Delfia assays. Did the recall rate change when assays changed but cut-off remained?
Response 4:
Thank you for this point. Between 1998 and 2006, the IRMA DPC (CLINITEST) assay was used, and in 2006, it was replaced by the Delfia® time-resolved fluorometry system. Although both platforms reported TSH values as serum-equivalent concentrations, methodological differences required a recalibration of the cut-off point from 20 to 15 µIU/mL in whole blood to maintain diagnostic equivalency (Page 3, line 87). The recall rate during the DPC period was 0.047% (95% CI: 3.6–6.2 per 10,000), which is comparable to the Delfia period recall rate of 0.05% (95% CI: 4.7–5.2 per 10,000). A clarifying sentence has been added (Page 6, line 167).
Comment 5:
L86: Please explain how whole blood values were corrected for hematocrit given differences in preterm vs term infants.
Response 5:
Thank you for this clarification request. We have added the conversion formula used for serum equivalence in the revised manuscript (Page 3, Line 104). A standardized correction factor is applied uniformly to both term and preterm infants, assuming an average hematocrit of 55%, whereby: 1 µU/mL in whole blood = 2.22 µU/mL in serum. While we acknowledge that hematocrit differs between term and preterm infants, a standardized correction factor based on an assumed hematocrit of 55% is applied uniformly across all gestational ages in routine screening practice. This ensures methodological consistency and comparability over time, though it may slightly overestimate serum-equivalent TSH in preterm infants.
Comment 6:
L87: Earlier, you state values reported as serum equivalents, but here refer to whole blood levels over 15 mIU/L. Is this a serum equivalent or whole blood?
Response 6:
Thank you for pointing this out. All TSH values reported in the manuscript are expressed as serum-equivalent concentrations. (Page 3, Line 104).
Comment 7:
L90: ‘If it remains’—does this refer to the average, median, highest, or another value of the repeats?
Response 7:
Thank you for the opportunity to clarify. If either of the duplicate retest results is greater than or equal to 15 µIU/mL in whole blood for Delfia, or 13 µIU/mL with GSP, the newborn is referred for clinical and biochemical evaluation (Page 3, line 87). We have clarified this statement in the Methods section, and Figures 1 and 2.
Comment 8:
L137: Please use appropriate statistical descriptions. The current format (128±73) implies a normal distribution, which may not be applicable.
Response 8:
Thank you for this important suggestion. While the sample size is smaller than the full screened population, each analytical period includes substantial number of screen-positive cases. Although TSH and T4 distributions are known to be right-skewed and are often reported using medians and interquartile ranges, the sample size is sufficient for the Central Limit Theorem to apply, permitting the use of mean and standard deviation as valid summary measures for comparative purposes between screening methods or calendar periods.
Comment 9:
L143: Recall rate is 0.05—please provide units (e.g., %). What was the positive predictive value for severe and mild cases?
Response 9:
Thank you for this request. We have clarified the recall rate and added positive predictive values to the revised manuscript (Page 6, Line 167).
- Recall rate: 0.05%.
- Positive predictive value (PPV):
o Severe CH cases: 58% (95% CI 53–63%)
o Mild CH cases: 14% (95% CI 12–16%)
Comment 10:
L189: The incidence of CH in preterm infants is higher than in term infants. Do you have separate incidence estimates for preterms?
Response 10:
Thank you for this question. Unfortunately, our screening database does not include the total number of preterm babies screened, only the number of preterm infants who screened positive. As a result, we cannot calculate an exact incidence rate specifically for the preterm population. We can, however, report that of the 1 594 confirmed CH cases, 186 (11.7%) occurred in infants born at ≤35 weeks and 102 (6.4%) in those born at 36–36⁶/₇ weeks. Given that preterm births represent approximately 8–10% of total live births in Chile, this suggests a substantially higher CH incidence in preterm versus term infants—consistent with literature—but an exact rate cannot be determined without the precise denominator. We have noted this limitation in the Discussion (Page 9, Line 257) and agree that the modest contribution of prematurity to overall case numbers makes it unlikely to fully account for the national upward trend.
Comment 11:
If there have been no changes in screening practice, population shifts or environmental factors may explain the increase (Albert et al., 2012; Chamot, 2025; Street et al., 2024). Do you have a hypothesis?
Response 11:
Thank you for this suggestion. As detailed in the Methods section, there have indeed been changes in screening methodology over time, including the transition from IRMA DPC to Delfia in 2006, and the implementation of the GSP platform in 2019. These changes were accompanied by adjustments in cut-off values, which may have contributed to the observed increase in CH incidence, particularly in mild cases.
In response to your comment, we have added a new paragraph to the Discussion (Page 9, Line 235). In addition to screening-related factors, we discuss the possible role of other factors. We agree that these hypotheses merit further investigation, and we have emphasized the need for future research integrating environmental, nutritional, and genetic data.